# Salivary Cystatin-L2-like of *Varroa destructor* Causes Lower Metabolism Activity and Abnormal Development in *Apis mellifera* Pupae

**DOI:** 10.3390/ani13233660

**Published:** 2023-11-27

**Authors:** He Zhou, Xinle Duan, Chaoxia Sun, Hongji Huang, Mei Yang, Shaokang Huang, Jianghong Li

**Affiliations:** 1College of Animal Sciences (College of Bee Science), Fujian Agriculture and Forestry University, Fuzhou 350002, China; hezhou@fafu.edu.cn (H.Z.); xinleduan@fafu.edu.cn (X.D.); chaoxiasun@fafu.edu.cn (C.S.); hjhuang@fafu.edu.cn (H.H.); yangmei@fafu.edu.cn (M.Y.); skhuang@fafu.edu.cn (S.H.); 2Fujian Honey Bee Biology Observation Station, Ministry of Agriculture and Rural Affairs, Fuzhou 350002, China

**Keywords:** salivary cystatin, *Varroa destructor*, manipulate, honeybee pupae, metabolism

## Abstract

**Simple Summary:**

*Varroa destructor* salivary secretion plays a vital role in mite–bee interactions. In this study, we found that the salivary cystatin gene was highly expressed in mites during the reproductive phase when they fed on pupal bees. Injection of prokaryotic-expressed cystatin into white-eyed pupal bees downregulated the metabolism of carbohydrates, fatty acids, and amino acids, and ATP production, which caused the pupal bees to fail to emerge and decreased the weight of newly emerged bees. Downregulation could save nutrients and energy for *V. destructor* to maximize its reproduction potential, implying that *Varroa destructor* could manipulate the metabolism of host bees through the injected salivary secretion. These results provide new insights into mite–bee interactions, providing a basis for *Varroa destructor* control in apiculture.

**Abstract:**

*Varroa destructor* injects a salivary secretion into honeybees during their feeding process. The salivary secretion plays a vital role in mite–bee interactions and is the main cause of honeybee illness. To determine the biological function of cystatin-L2-like, one of the components of *V. destructor* salivary secretion, its gene expression in mites during the reproductive phase and dispersal phase was quantified using RT-qPCR, respectively. Moreover, the *E. coli*-expressed and -purified cystatin was injected into the white-eyed honeybee pupae, and its effects on the survival, the weight of the newly emerged bee, and the transcriptome were determined. The results showed that cystatin was significantly upregulated in mites during the reproductive phase. Cystatin significantly shortened the lifespan of pupae and decreased the weight of the newly emerged bees. Transcriptome sequencing showed that cystatin upregulated 1496 genes and downregulated 1483 genes in pupae. These genes were mainly enriched in ATP synthesis, the mitochondrial respiratory chain, and cuticle structure and function. Cystatin comprehensively downregulated the metabolism of carbohydrates, fatty acids, and amino acids, and energy production in the pupae. The downregulation of metabolic activity could save more nutrients and energy for *V. destructor*, helping it to maximize its reproduction potential, implying that the mite could manipulate the metabolism of host bees through the injected salivary secretion. The results provide new insights into mite–bee interactions, providing a basis for related studies and applications.

## 1. Introduction

Honeybees can produce many nutritious, healthy, and valuable bee products for human beings [1]. Moreover, as an efficient pollinator, the honeybee is indispensable for maintaining global ecological balance and agricultural production. Previous reports showed that in the USA alone, pollination results in USD 16 billion annually with USD 12 billion attributable solely to the accessibility of honeybees [2,3]. However, 20.9% of honeybee colonies were lost in the winter of 2016/2017, based on the data from 425,762 colonies in 30 countries [4]. The loss was mainly caused by multiple biotic or abiotic factors, or their combined effects [5,6,7,8]. The decrease in honeybee colony numbers cannot meet the requirements of crop production in agriculture, which threatens the world’s food security and has caused concern all over the world for the past few decades.

Among the multiples biotic factors threatening honeybee survival, *Varroa destructor* is always the greatest threat to European honeybees (*Apis mellifera* L.) worldwide [9,10]. With the aid of the international commercial trade of honeybees and bee products, *V. destructor*, as an ectoparasitic mite native to the Asian honeybee (*Apis cerana*), spread to the European honeybee probably around the 1950s in Asia, and then rapidly spread in the 1970s in Europe and the 1980s in America [10]. Parasitic mites could cause the collapse of most honeybee colonies if left untreated. Given the scale and global distribution of *A. mellifera*, *V. destructor* has been linked to a worldwide decline in honeybee health and has had a devastating impact on apiculture [11,12,13]. Despite the great progress achieved in research and application, *V. destructor* currently remains a major beekeeping issue throughout the world.

The life cycle of *V. destructor* can be divided into the reproductive phase, which takes place inside honeybee brood cells, where a foundress mite rears her young, and the dispersal phase (or phoretic phase), in which mature female mites travel and feed on adult bees [14]. The mites in the reproductive phase stay inside the capped cell of the honeybee brood, which protects them from contact with all kinds of miticides used in the beekeeping industry. A foundress mite can have up to seven reproductive cycles and lay up to 30 eggs in her life under laboratory conditions [15], and can produce ~5–10 mature daughters in a worker brood or ~10–17 mature daughters in a drone brood [10]. The reproduction and growth of mites completely rely on feeding on the hemolymph and the fat body of host bees [16]. In addition, *V. destructor* is a vector for various honeybee viruses [17,18,19]. Therefore, the mite-feeding activity also increases the transmission and proliferation of honeybee viruses [20,21]. A substantial loss of nutrients and the amplified virus cause the brood to fail to emerge or become deformed. In adult bees, *V. destructor* can decrease the body weight and water content of young emerging bees [22], alter the flying, homing, and orientation abilities of foragers [23], downregulate the expression of immune genes in emerging infested adults [24,25,26], and decrease the lifespan [27,28].

*V. destructor* feeds on the bodily fluids of brood or adult bees through a wound in the cuticle, which allows food uptake by the mother mite and its progeny [29,30]. Later research found that *V. destructor* feeds primarily on honeybee fat body tissues [16]. The feeding activity was associated with saliva injection. Therefore, the components and their specific functions in salivary secretions have attracted significant attention. The mite’s salivary secretion can damage the hemocytes and suppress their ability to extend pseudopods and form aggregates, implying their ability to suppress hemocyte-mediated wound healing and plugging responses of the host [31]. A toxic protein, also responsible for the elevated DWV titers and the subsequent development of deformed-wing adult bees, was discovered from salivary secretion [32]. Salivary chitinase was found to be involved in keeping the feeding wound open and preventing host infection by opportunistic pathogens [33]. The esterases from the mite’s salivary secretion might be related to its nutritional absorption [34]. Moreover, both the whole salivary secretion and the specific protein in the secretion were found to be involved in the immune response of host bees [35,36]. All these reports show the important biological functions of the *V. destructor* salivary secretion. So far, over 300 proteins have been determined from the salivary secretome via nano-liquid chromatography coupled with tandem mass spectrometry [37]. Except for a few proteins with their functions determined, the functions of the remaining proteins still need to be determined. Genomic sequencing of *V. destructor* showed that there were six cystatin-related mRNAs among the nine salivary-related mRNAs, implying the vital roles of cystatin in mites. To determine the biological function of cystatin, which was not previously reported, *V. destructor* salivary cystatin-L2-like (LOC111252001, hereafter referred to as cystatin) was expressed and purified from the *E. coli*. Then, the purified protein was injected into white-eyed pupae, and its effects on honeybees were determined in detail.

## 2. Materials and Methods

### 2.1. Honeybees

The honeybees of *A. mellifera* used in the experiment were collected from the colonies kept in the experimental apiary of the College of Animal Sciences (College of Bee Science), Fujian Agriculture and Forestry University. Ten colonies were intentionally kept without mite control, from which the mites in the study were collected.

### 2.2. Acquisition of V. destructor

To determine the difference in gene expression of cystatin in Varroa mites in the reproductive phase and the dispersal phase, three colonies with *V. destructor* infection were selected for mite sampling. Specifically, for collecting the Varroa mites of the reproductive phase, one frame with elder brood and capped brood was removed from each colony, the cell’s cap was opened with forceps and the white-eyed pupae were carefully pulled out of the cell; the mite on the white-eyed pupae was collected, and around 30 mites per colony were pooled into a 1.5 mL Eppendorf tube (Axygen, Tewksbury, MA, USA) as one sample. The Varroa mites of the dispersal phase were collected according to a previous report [38,39], and 30 mites were similarly pooled as one sample per colony in the tube. Three samples of mites of the reproductive phase and mites of the dispersal phase from three colonies, respectively, were prepared and stored at −80 °C immediately.

### 2.3. RNA Extraction and cDNA Synthesis

TRIzol^®^ Reagent (Invitrogen, Carlsbad, CA, USA) was used for extracting RNA from these stored mites as described by the protocol. After inspection by the NanoDrop One (Thermo Fisher Scientific™, Waltham, MA, USA), the RNA was stored at −80 °C. cDNA was synthesized from these RNA samples using a reagent kit according to manual instructions (Hifair^®^ 1st Strand cDNA Synthesis SuperMix for qPCR, Yeasen Biotech Co., Ltd., Shanghai, China), and then stored at −20 °C for use.

### 2.4. Cystatin Gene Cloning, Expression, and Purification

To comprehensively determine the protein’s function on honeybees, a prokaryotic expression system was set up for obtaining the cystatin in the study. In detail, the salivary cystatin-L2-like gene of *V. destructor* has four isoforms. By comparing and analyzing the sequence of the four isoforms, we designed a cystatin coding DNA sequence containing the conserved sequence of the four isoforms by removing the variant part, and adjusting a part of the nucleic acid base based on the code usage bias of *E. coli* so that the protein could be efficiently expressed in *E. coli*. Restriction sites of NdeI (CATATG) and XhoI (CTCGAG) were added to the 5′- end and 3′- end of the sequence, respectively. The designed sequence was artificially de novo synthesized by Sangon Biotech Co., Ltd., Shanghai. For the optimized amino acid sequence and DNA sequence of cystatin, refer to Supplementary File S1. After digestion by NdeI and XhoI, the DNA fragments and the plasmid DNA of pET-22b (+) (Novagen, Madison, WI, USA) were connected using ligase and then transformed into the competent cell of *E. coli*, DH 5α. The recombinant plasmid was screened and transformed into an *E. coli* of BL21 (DE3) to induce the expression of the objective protein. After being verified by Western blot using the anti-His body as the primary antibody, the expressed protein was purified by Ni-NTA (His-tag Affinity). The concentration of the purified protein was determined by the Lowry Protein Assay Kit (Sangon Biotech Co., Ltd., Shanghai, China) and then stored at −80 °C.

### 2.5. Cystatin Injection into Honeybee Pupae

We tested the injection of 400 ng of cystatin into pupal bees, referring to a previous report of injecting Varroa mite saliva into both *A. mellifera* and *A. cerana* [37]. However, all the test pupal bees died quickly. Then, three rounds of two-times-diluted cystatin were tested and showed different lethal rate levels. Therefore, the three series concentrations (200, 100, and 50 ng/µL) of the purified cystatin in PBS were selected for the injection of honeybee pupae in this study. Specifically, frames with the elder brood and capped brood were removed from the colony, and the capped brood was opened using forceps. The white-eye pupae were carefully blown out of the cells with a rubber pipette bulb. These pupae were divided into five groups, which were, respectively, injected with 1 µL of the series-diluted cystatin (Groups: Cys200, Cys100, and Cys50, corresponding to the concentrations of 200, 100, and 50 ng/µL, respectively), PBS by a microinjector (Group: PBS), and nothing as the control (Group: CK). A total of 96 pupae on four 24-well plates for each group were prepared. After injection, these pupae were kept in an incubator (Thermo Fisher Scientific Co., Ltd., Shanghai, China) at 34.5 °C, 75% RH, and were photographed every day. The developmental state of these pupae was determined by their appearance and change in their body color over time. Pupae with no appearance and body color change for two consecutive days were designated as growth arrest and finally counted as dead in the survival curve. Each of the newly emerged bees was weighed with an analytical balance (XPR226DRQ, Mettler Toledo, Shanghai, China) to analyze the effect of cystatin on honeybee body weight.

### 2.6. Transcriptome Sequencing

To determine the effect of cystatin on the gene expression of honeybee pupae, the pupae 3 days post injection from groups Cys200 and PBS were selected for transcriptome sequencing. In detail, 10 pupae were pooled as one sample from each 24-well plate. Three samples (repeats) were prepared for groups Cys200 and PBS, respectively. RNA was extracted from these samples as described above. RNA quality and quantitation were determined by the Qubit™ RNA HS Assay Kit according to the manuals (Life Technologies, Gaithersburg, MD, USA). Sequencing libraries were generated using Hieff NGS™ MaxUp Dual-mode mRNA Library Prep Kit for Illumina^®^ (Yeasen Biotech Co., Ltd., Shanghai, China) according to the instructions. The libraries were subjected to an Illumina Hiseq™ platform for generating 150 bp paired-end raw reads, from which clean data (clean reads) were obtained by removing the adapter containing reads, ploy-N reads, and low-quality reads using Trimmomatic v0.36 [40]. The clean reads were then aligned to the genome of honeybee *A. mellifera* using Hisat2 v2.1.0 [41]. The mapped reads were further assembled by StringTie v1.3.3b [42] referring to the genome of honeybee *A. mellifera*.

### 2.7. Analysis of Differentially Expressed Genes

The number of reads mapped to each gene was counted using FeatureCounts v1.5.0 [43]. The expression of each gene was analyzed using the RPKM method (reads per kilobase of transcript per million reads mapped). The FPKM (Fragments Per Kilobase of exon model per Million fragments mapped) was introduced to remove the deviation caused by the gene length on expression analysis. Differentially expressed genes between the cys200 and PBS groups were screened using the DESeq2 R package v1.12.4 [44]. The optimized Q-value from the *p*-values using Benjamini and Hochberg’s approach was selected for controlling the false discovery rate. In addition, genes with a Q-value < 0.05 were assigned as differentially expressed.

### 2.8. GO and KEGG Enrichment Analysis of Differentially Expressed Genes

GO enrichment was performed using topGO R package v2.24.0, and KEGG pathway enrichment was implemented using the cluster Profiler R package v3.0.5 [45], in which gene length bias was corrected. GO terms with a Q-value less than 0.05 were considered significantly enriched from these differentially expressed genes.

### 2.9. qRT-PCR Validation

qRT-PCR was used to determine the difference in gene expression of cystatin in Varroa mites in the reproductive phase and the dispersal phase. The synthesized cDNA from the extracted RNA of *V. destructor* was used as a template. A pair of cystatin-specific primers was used in the PCR (Appendix A), and the *V. destructor* actin gene was used as a reference [46]. The reactions were set up using SYBR Green Master Mix (Yeasen Biotech Co., Ltd., Shanghai, China), and then ran in an ABI QuantStudio 6 Flex System (Thermo Fisher Scientific, Waltham, MA, USA). The running program was as follows: denaturation at 95 °C for 3 min, then 45 cycles of denaturation at 95 °C for 15 s, annealing at 60 °C for 20 s, and extension at 72 °C for 20 s.

Moreover, 16 genes based on the KEGG enrichment pathway (*LOC552671* and *LOC727599* from oxidative phosphorylation, *LOC724724*, *LOC550667*, and *LOC725325* from carbon metabolism, *LOC412815* and *LOC410241* from AMPK signaling, *LOC552328* and *LOC552286* from insulin signaling, *LOC724389*, *LOC552209*, and *LOC413190* from calcium signaling, etc.) were quantified via qRT-PCR to validate the reliability of the transcriptome sequencing using the specific primers (Appendix A). The same PCR program and an ABI QuantStudio 6 Flex System, as described above, were utilized. The *β-actin* was used as a reference [47].

### 2.10. Statistical Analysis

The gene expression was calculated using the 2^−ΔΔCt^ equation. The difference in the expression of the cystatin gene between the mites in the reproductive phase and dispersal phase, and the 16 genes for validating the reliability of the result of transcriptome sequencing between the cys200 and PBS groups were analyzed via an independent *t*-test. The differences in the weight of newly emerged bees from the five test groups were analyzed using one-way ANOVA, followed by Tukey’s post hoc multiple comparisons test. All data in the study were expressed as the mean ± SD (standard deviation). The survival curve was constructed based on the data of daily dead pupae, and the difference between the test groups was analyzed using the log-rank (Mantel–Cox) test. All the analyses were performed in GraphPad Prism (Version 8.0, Graph Pad Software Inc., San Diego, CA, USA), and *p* < 0.05 was set as the significance.

## 3. Results

### 3.1. Cystatin Was Highly Expressed in V. destructor in the Reproductive Phase

By quantifying the gene expression of cystatin in *V. destructor* using qRT-PCR, we found that the gene of cystatin was significantly more highly expressed in the mites in the reproductive phase than in the mites in the dispersal phase (Independent *t*-test: *t* = 5.629, *df* = 4, *p* = 0.0049) (Figure 1). This implied that the salivary cystatin secretion was more active in Varroa mites in the reproductive phase when they were feeding on honeybee pupae.

### 3.2. Cloning, Expression, and Cystatin Purification

The de novo synthesized DNA coding cystatin was restricted with NdeI and XhoI and then ligated with the expression vector of pET-22b (+). The recombinant plasmid was screened by agarose gel electrophoresis of plasmid DNA digested with NdeI and XhoI, as shown in Figure 2A. After the inserted DNA sequence was further verified by double-strand sequencing, the recombinant plasmid was transformed into the *E. coli* of BL21 (DE3) to induce the expression of the cystatin protein. A protein band with a molecular weight similar to the predicted cystatin was successfully induced to express in the precipitation of cell lysis (Figure 2B). Western blot further validated that the protein was His-fusion cystatin protein (Figure 2C). The cystatin protein was successfully purified with Ni-NTA (His-tag Affinity) (Figure 2D). The original images for blots/gels refer to File S2. The concentration of the purified protein was 0.85 mg/mL.

### 3.3. Cystatin Decreased the Survival of Honeybee Pupae

The survival rates of the pupal bees from the five groups were 98.78, 86.72, 34.40, 12.16, and 9.86% for the groups CK, PBS, Cys50, Cys100, and Cys200, respectively (Figure 3A). The survival rate decreased with the increase in the dosage of cystatin injected. The lethal effect was dosage-dependent. Therefore, cystatin significantly decreased the survivorship of honeybee pupae (Log-rank test, *χ*^2^ = 221.5, *p* < 0.001). Most of the pupal bees injected with cystatin died before emergence, with their cuticle turning dark (Figure 3B).

### 3.4. Cystatin Decreased the Weight of Newly Emerged Honeybees

By analyzing the weights of newly emerged honeybees, we found that the weights of these test bees among the five groups were significantly different (one-way ANOVA, *F* = 26.38, *p* < 0.001). Tukey’s multiple comparisons tests showed that the weights of newly emerged honeybees from group CK (111.3 ± 0.8947 mg) and PBS (111.4 ± 1.338 mg) were not significantly different (*p* > 0.9999). However, the weights of the bees from all three cystatin-injected groups (for Cys50: 96.99 ± 2.151 mg; for Cys100: 91.70 ± 3.035 mg; and for Cys200: 83.84 ± 7.067 mg) were significantly lower than the bees from the CK and PBS groups (for the three cystatin-injected groups vs. CK, *p* < 0.001; for the three cystatin-injected groups vs. PBS, *p* < 0.001). Moreover, the weights of honeybees among the three cystatin-injected groups were also different (for Cys50 vs. Cys100, *p* = 0.6804; for Cys100 vs. Cys200, *p* = 0.5705; for Cys50 vs. Cys200, *p* = 0.0273) (Figure 4). Thereby, the weight-decreasing effect of cystatin on the newly emerged honeybee was also dosage-dependent.

### 3.5. Transcriptome Sequencing and Differentially Expressed Genes

For determining the comprehensive effect of cystatin on pupal bees, the pupae from the PBS and Cys200 groups were subjected to transcriptome sequencing. We finally obtained clean reads ranging from 64,357,262 to 95,132,698 from the six libraries, with an average Q30 value of 94.16. A total of 13,173 unigenes were assembled using StringTie (v1.3.3b) from the clean reads mapped to the *A. mellifera* genome by Hisat2. The original sequencing data of the six libraries of pupae from the group PBS and Cys200 were deposited in the Sequence Read Archives (SRAs) under the BioProject of PRJNA1029743 at NCBI.

Differentially expressed genes in pupal bees between the PBS and Cys200 were analyzed using the DESeq2 R package. Among the 13,173 unigenes assembled, cystatin caused 1496 genes to be upregulated, and 1483 genes to be downregulated (Figure 5A). The differentially expressed genes refer to Appendix A. Cystatin actively interfered with gene expression in honeybee pupae. Clustering analysis showed that the gene expression from the three samples of PBS groups could be clustered into one branch, while the gene expression from the three samples of Cys200 groups could be clustered into another branch (Figure 5B). Such a similarity in gene expression profiles of each group implied a good repeatability of tested samples and the following sequencing.

### 3.6. GO Enrichment

GO enrichment analysis showed that the differentially expressed genes were enriched in 133 items of biological process, 63 items of cellular components, and 48 items of molecular function, respectively (Appendix A). The most enriched items in the biological process were respiratory and energy metabolism-related such as the oxidation–reduction process (GO: 0055114), the generation of precursor metabolites and energy (GO: 0006091), drug metabolic process (GO: 0017144), small-molecule metabolic process (GO: 0044281), energy derivation by the oxidation of organic compounds (GO: 0015980), ATP metabolic process (GO: 0046034), ATP synthesis coupled electron transport (GO: 0042773), mitochondrial ATP synthesis coupled electron transport (GO: 0042775), electron transport chain (GO: 0022900), respiratory electron transport chain (GO: 0022904), etc. The most enriched items in the cellular component were mainly related to the mitochondrial respiratory chain and extracellular region, such as extracellular region (GO: 0005576), extracellular region part (GO: 0044421), respiratory chain complex (GO: 0098803), respiratory chain (GO: 0070469), extracellular space (GO: 0005615), mitochondrial respiratory chain (GO: 0005746), inner mitochondrial membrane protein complex (GO: 0098800), oxidoreductase complex (GO: 1990204), mitochondrial membrane part (GO: 0044455), extracellular matrix (GO: 0031012), etc. The most enriched items in molecular function were oxidoreductase activity and cuticle-related functions, such as oxidoreductase activity (GO: 0016491), serine-type peptidase activity (GO: 0008236), structural constituent of the chitin-based cuticle (GO: 0005214), structural constituent of the cuticle (GO: 0042302), transmembrane transporter activity (GO: 0022857), serine hydrolase activity (GO: 0017171), proton transmembrane transporter activity (GO: 0015078), ion transmembrane transporter activity (GO: 0015075), inorganic molecular entity transmembrane transporter activity (GO: 0015318), serine-type endopeptidase activity (GO: 0004252), etc. (Figure 6). Thereby, these differentially expressed genes caused by the cystatin were mainly enriched in the items of ATP synthesis, mitochondrial respiratory chain, cuticle structure and function, etc.

### 3.7. KEGG Enrichment

KEGG enrichment showed that a total of 23 items were significantly enriched (Appendix A). Among them, the most enriched functions were oxidative phosphorylation (ko00190), carbon metabolism (ko01200), ECM–receptor interaction (ko04512), fatty acid metabolism (ko01212), cardiac muscle contraction (ko04260), citrate cycle (TCA cycle) (ko00020), biosynthesis of amino acids (ko01230), starch and sucrose metabolism (ko00500), glyoxylate and dicarboxylate metabolism (ko00630), and glycolysis/gluconeogenesis (ko00010). In addition, the AMPK signal pathway, ECM–receptor interaction, and steroid hormone biosynthesis were also enriched (Figure 7). Such a result showed that the cystatin significantly disturbed the process of energy metabolism. Furthermore, the signal transduction and steroid hormone biosynthesis process were also affected.

To further reveal the relation between the KEGG-enriched functions and differentially expressed genes, an interaction network of the top 10 enriched functions and involved genes was constructed. The result showed that nine functions involving the metabolism of carbon, fatty acids, amino acids, starch, sucrose, and energy production-related processes such as oxidative phosphorylation and the citrate cycle (TCA cycle) were well interconnected. Only ECM–receptor interaction was not included in the network. Moreover, most of the circle node was green in color, which means the genes involved in the enriched functions were downregulated (Figure 8). Such a result showed that the cystatin comprehensively downregulated the metabolism of carbohydrates, fatty acids, amino acids, and the energy production of honeybee pupae.

### 3.8. Validation

Considering that the downregulated genes were well enriched in KEGG analysis, 16 downregulated genes from the KEGG-enriched pathways were selected for the validation of their expression by qRT-PCR using gene specific primers (Appendix A). The result showed that all 16 genes had a single peak of dissolution curve. Their expression in the pupae of the Cys200 group was about 10–50% of the expression in the pupae of the PBS group (Figure 9). Such a downregulated expression determined by qRT-PCR was similar to the expression determined via RNA-seq analysis (*t*-test, for all the sixteen genes, *p* > 0.05), which validated the accuracy of the RNA-seq results.

## 4. Discussion

As an obligate parasite of honeybees, *V. destructor* completely relies on feeding on the hemolymph and fat body of honeybees for its survival and reproduction [16]. The salivary secretion of *V. destructor* plays a vital role in this process. Thereby, research on the component and function of the specific protein in salivary secretion is necessary for understanding the interaction between *V. destructor* and the honeybee host [37]. Previous research has determined some proteins and their function based on salivary secretion [31,32,33,34,35,36,37]. In this study, based on the genomic sequence, we found that cystatin was one of the main components in the salivary secretion of *V. destructor*, with undetermined functions. To explore it, we expressed the cystatin from *E. coli* and injected it into white-eyed pupal bees. Its effects on pupal bee survival, the weight of newly emerged bees, and the transcriptome response were investigated in detail. The results showed that cystatin was lethal to pupae in a dose-dependent manner, could decrease the weight of newly emerged bees, and could fundamentally alter the host bee’s gene expression profile. Moreover, a previous report showed that the injection of 432 ng of Varroa mite saliva caused deformed wings in newly emerged bees of *A mellifera*, which was equal to the symptoms caused by two Varroa mites in the lab [37]. In this study, injection of the lowest dosage of 50 ng cystatin could cause over 50% of pupal bees to die, exhibiting much stronger toxicity to honeybees than the Varroa mite itself or the Varroa mite’s saliva. To our knowledge, this is a new protein component in the salivary secretion of *V. destructor* with effects on host bees determined except for in the previous reports [31,32,33,34]. Such a result deepens our knowledge of the salivary secretion of *V. destructor* and the interaction between *V. destructor* and honeybees.

The life of *V. destructor* on the honeybee host involves two distinct stages: the reproductive phase on pupae and the dispersal phase on adult bees [14]. The mites in the reproductive phase ingest as much hemolymph and fat body as possible to produce offspring [16]. However, the mites in the dispersal phase feed on adult bees just to maintain their survival and pursue the opportunity for the next round of brood infection. The amount of hemolymph and fat body ingested by mites in the dispersal phase is reasonably less than mites in the reproductive phase. Our results showed that the gene expression level of cystatin was significantly increased in mites in the reproductive phase compared to mites in the dispersal phase (Figure 1). The secretion of salivary cystatin in *V. destructor* was positively related to its feeding activity and nutrient ingestion in the reproductive phase.

To determine the effect of cystatin on honeybees, the *E. coli*-expressed and purified cystatin was injected into the white-eyed pupae, and the growth and development of the treated bees were determined. The result showed that the survival of cystatin-injected pupae was significantly decreased compared to pupae from the control group and that most of the pupae died before emergence (Figure 3). Moreover, cystatin treatment also decreased the weight of newly emerged bees (Figure 4). Such an effect of cystatin was similar to the effect of *V. destructor* on honeybees, which implied that the mite infection of honeybees might in great part be caused by the salivary protein.

Transcriptome analysis can provide a comprehensive analysis of gene expression changes when an organism is subject to all kinds of stress. Our transcriptome analysis showed that nearly 3000 genes were differentially expressed in cystatin-injected pupae compared with PBS-injected pupae (Figure 5). This change in the expression level of so many genes severely disrupted the normal growth and development of honeybees, causing fewer pupae to emerge and significantly decreasing the weight of newly emerged bees (Figure 3 and Figure 4). GO analysis showed that the differentially expressed genes were mainly enriched in the items of ATP synthesis, mitochondrial respiratory chain, cuticle structure and function, etc. (Figure 6). The metabolism of carbon, fatty acids, amino acids, starch, and sucrose, and energy production-related processes such as oxidative phosphorylation and the citrate cycle (TCA cycle) were well enriched in KEGG (Figure 7). Moreover, most genes involved were downregulated (Figure 8). Such results suggested that cystatin could slow down the utilization of nutrients and ATP production in pupal bees. When *V. destructor* in the reproductive phase sucks hemolymph and fat body, this causes a decrease in the nutrient levels of the pupal bee. Thereby, the downregulation of nutrient utilization might be some kind of response of the host pupal bees to the nutrient deprivation caused by *V. destructor*. In this study, cystatin injection did not deprive the pupal bees of nutrients. However, the test bees still downregulated gene expression involving nutrient utilization and ATP production. This reflected how the honeybee, as one side of the army race, evolved some mechanism at the gene expression level to cope with the feeding of *V. destructor* through sensing the injected salivary secretion.

Moreover, cuticle structure and function were well enriched from the differentially expressed genes in this study. This might be caused by the downregulation of pupal bee nutrient utilization because the nutrient level fundamentally determines insect cuticle development [48,49].

From the view of *V. destructor*, the downregulation of nutrient utilization and ATP production in pupal bees hold additional significance. On the one hand, the downregulation of nutrient utilization implied that fewer nutrients were activated to build the pupae tissues, which guaranteed that *V. destructor* could obtain more nutrients from the hemolymph and fat body through their feeding activity. On the other hand, any biochemical process, including development, requires the consumption of a large amount of ATP. Downregulated ATP production could slow down the pupae’s development process. Previous reports have shown that only honeybee larvae within 18 h (in workers) and 36 h (in drones), respectively, after cell capping, were able to stimulate the mite’s oogenesis [50]. Thereby, the slower development process of pupal bees caused by salivary cystatin could provide *V. destructor* with a longer time window for oogenesis and egg laying. From this point, *V. destructor* could manipulate (slow down) the development process of the host pupal bee to maximize its reproduction potential by injecting the salivary secretion.

Cysteine is a common amino acid, and the disulfide bond formed between two cysteine residues is essential for forming the tertiary and quaternary structure of all kinds of functional proteins. As holometabolous insects, brood bees store large amounts of nutrients in the fat body, which are completely disrupted for the purposes of pupae tissue building in metamorphosis [51]. The disruption of the stored protein requires the deconstruction of its tertiary and quaternary structures, which involves the function of cysteine proteinases such as caspases [52]. Insect metamorphosis is triggered by a pulse of steroid hormones [53]. This study also revealed that cystatin induced the alteration in the biosynthesis of steroid hormones (as shown in Figure 7). Previous reports showed that a Drosophila caspase, DRONC, an effector of steroid-mediated apoptosis, could be induced by steroids, and that it is required for insect metamorphosis [54,55]. Thereby, cysteine proteinases such as caspases positively participate in insect metamorphosis. However, as an inhibitor of cysteine proteinase, cystatin could inhibit the activity of caspase or other cysteine proteinases, which might cause the storage protein disruption process in the brood, and retard or even block the building of pupae tissues in metamorphosis, meaning that the nutrients stored in the brood are unable to be disrupted to rebuild pupae tissues in metamorphosis development. In addition to the above transcriptome data of decreased energy metabolism, once again we can conclude that *V. destructor* could intervene in the normal process of metamorphosis through blocking the disruption of storage protein in the brood and the construction of pupae tissues. This leads to nutrient and energy conservation, meaning *V. destructor* can maximize its reproduction potential by injecting salivary cystatin into host bees in the feeding process.

*V. destructor* could act as a vector and disseminator of honeybee viruses between and within bee colonies, and as an activator of virus multiplication in infected individuals. The deformed wing virus (DWV) in *V. destructor*-infested colonies is now considered to be one of the key contributors to the final collapse [56]. Following the infection of *V. destructor*, the amount of DWV is often sharply increased in individual honeybees [16,57]. For this reason, *V. destructor* has dramatically altered DWV epidemiology worldwide [58]. To determine whether the DWV increase was caused by the salivary cystatin, we quantified the amount of DWV in the test honeybee using qRT-PCR. The result showed that the DWV amount in honeybees between the cystatin-injected bees and the PBS-injected bees has no significant difference (Cystatin does not affect DWV multiplication. When *V. destructor* feeds on honeybees, DWV in the salivary secretion can be simultaneously injected into the host bee, which might, in part, account for the sharp increase in DWV [37,59]. In addition, the previous report showed that some protein components in *V. destructor* salivary secretion could activate DWV replication in the host bee, causing the DWV to increase too [32]. In our study, only the purified cystatin not containing DWV or other DWV activators was injected into the host bee; thus, the non-increasing effect on DWV is understandable.

## 5. Conclusions

Salivary secretion mediated the interaction between *V. destructor* and its host bees. The biological functions of specific salivary secretion proteins that are involved in this interaction have been continuously studied in the field over a long time period. In this study, cystatin was determined to be significantly highly expressed in the reproductive-stage of *V. destructor* and was verified to be virulent to pupal bees, shortening their life span and decreasing the weight of newly emerged bees. Cystatin changed the expression of nearly 3000 genes of pupal bees. These genes were mainly enriched in the downregulation of metabolic activity, affecting sugars, fats, and amino acids in the host pupal bees. Such downregulation implied that *V. destructor* could manipulate the metabolism of pupal bees through injected salivary secretion to save more nutrients and energy for self-reproduction. This result introduces new data on the interaction between *V. destructor* and honeybees, and provides an experimental basis for *V. destructor* control in apiculture.

## Figures and Tables

**Figure 1 animals-13-03660-f001:**
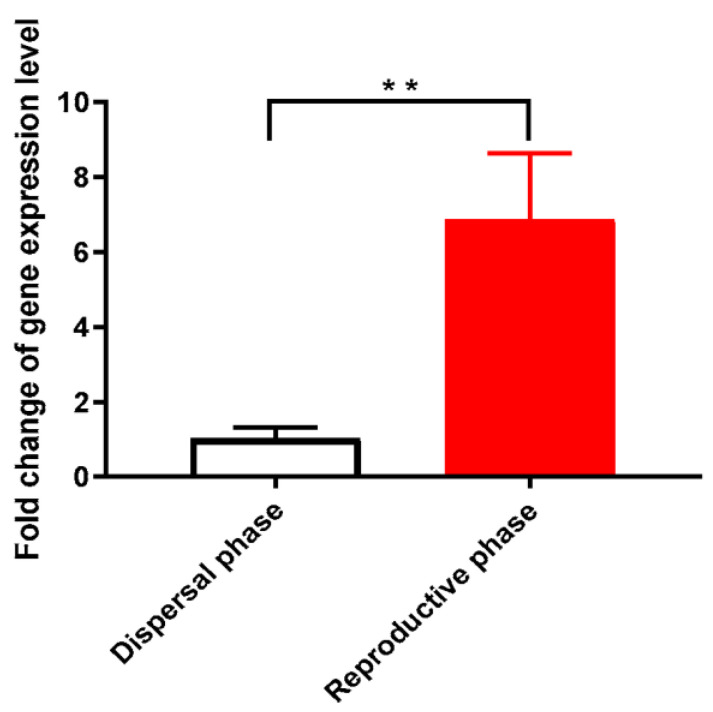
The expression of the cystatin gene in mites in the reproductive phase and dispersal phase. The gene expression of cystatin in the Varroa mites in the reproductive phase was significantly higher than the expression in the mites in the dispersal phase (independent *t*-test: *t* = 5.629, *df* = 4, *p* = 0.0049). Note: ** means *p* < 0.01, same as below.

**Figure 2 animals-13-03660-f002:**
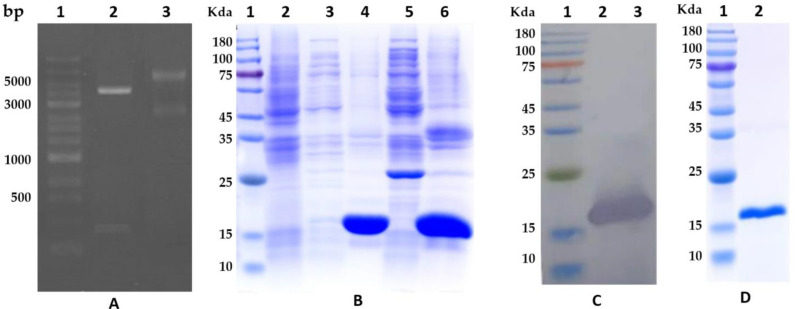
Cloning, expression, and purification of cystatin from *E. coli*. (**A**): Screening of the recombinant plasmid. Lane 1, DNA marker; Lane 2, the recombinant plasmid DNA restricted with NdeI and XhoI; Lane 3, the non-restricted plasmid DNA. (**B**): *E. coli* expression of cystatin. Lane 1, protein marker; Lane 2, non-induced lysis sample of the *E. coli*; Lane 3, supernatant of the lysis from *E. coli* induced at 20 °C; Lane 4, precipitation of the lysis from *E. coli* induced at 20 °C; Lane 5, supernatant of the lysis from *E. coli* induced at 37 °C; Lane 6, precipitation of the lysis from *E. coli* induced at 37 °C. A 15 KDa protein similar to the predicted cystatin was over-expressed in the precipitation of *E. coli* lysis. (**C**): Western blot result of the protein from the precipitation of *E. coli* lysis confirms the protein was a His-fusion cystatin protein. (**D**): The purified His-fusion cystatin.

**Figure 3 animals-13-03660-f003:**
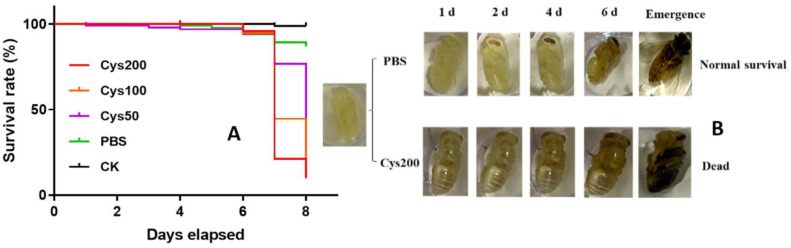
Effect of cystatin on the survival of honeybee pupae. (**A**): survival curve of series-diluted cystatin on honeybee pupae. CK: control group (black line); PBS: pupae injected with PBS (green line); Cys50: pupae injected with 50 ng/µL of cystatin in PBS (purple line); Cys100: pupae injected with 100 ng/µL of cystatin in PBS (yellow line); Cys200: pupae injected with 200 ng/µL of cystatin in PBS (red line); survival curves show cystatin significantly decreased the survivorship of honeybee pupae (log-rank test, *χ*^2^ = 221.5, *p* < 0.001). (**B**): development state of pupal bees injected with PBS and 200 ng/µL of cystatin, respectively. Most of the cystatin-injected pupal bees died before emergence, and their cuticles turned dark.

**Figure 4 animals-13-03660-f004:**
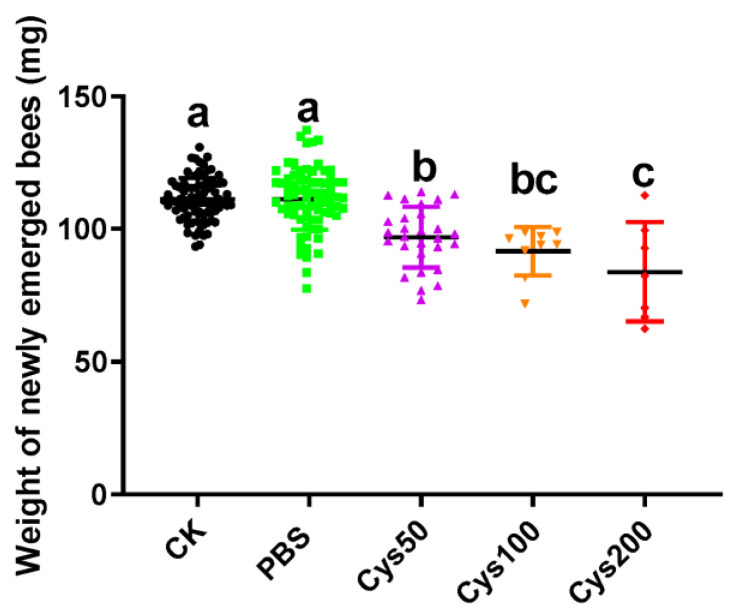
Effect of cystatin on the weight of newly emerged bees. CK: control group (black dot); PBS: PBS-injected group (green square). Cys50: pupae injected with 50 ng/µL of cystatin in PBS (purple triangle); Cys100: pupae injected with 100 ng/µL of cystatin in PBS (yellow inverted triangle); Cys200: pupae injected with 200 ng/µL of cystatin in PBS (red diamond); cystatin significantly decreased the weight of newly emerged adults (one-way ANOVA, *F* = 26.38, *p* < 0.001). The weight-decreasing effect of cystatin on the newly emerged honeybee was dosage-dependent. Different letters above each bar indicate statistically significant differences (*p* < 0.05).

**Figure 5 animals-13-03660-f005:**
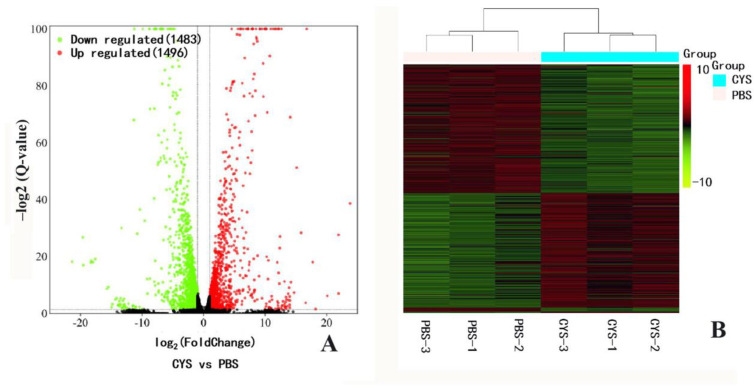
Volcano plot (**A**) and hierarchical clustering (**B**) of differentially expressed unigenes (DEGs) between the PBS- and cystatin-injected honeybee pupae. The expression level for each unigene was shown in the volcano plot. The *X*-axis shows a fold change of the gene between PBS- and cystatin-injected honeybee pupae. The *Y*-axis means Log10 of the fold change of the unigene. The significant DEGs were considered as fold change ≥ 1 and Q-value ≤ 0.05. Clustering analysis of DEGs showed that their expression mode in the three PBS samples could be clustered in one branch, while that in the three cystatin-injected samples could be clustered in another branch. The similar expression profiles of DEGs in the three samples of both PBS and cystatin groups implied good test repeatability.

**Figure 6 animals-13-03660-f006:**
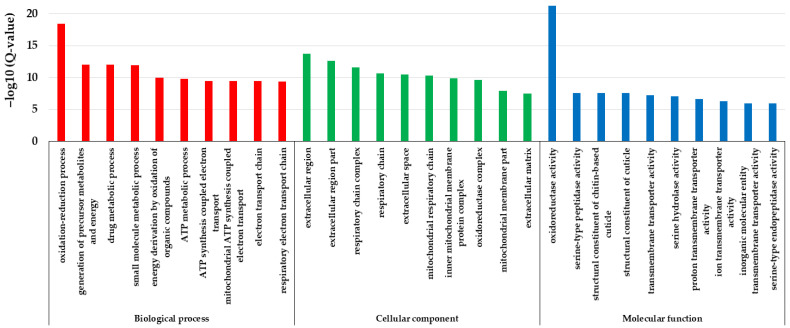
GO enrichment of the differentially expressed genes in pupal bees caused by cystatin. The items of ATP synthesis, mitochondrial respiratory chain, cuticle structure and function, etc., were enriched.

**Figure 7 animals-13-03660-f007:**
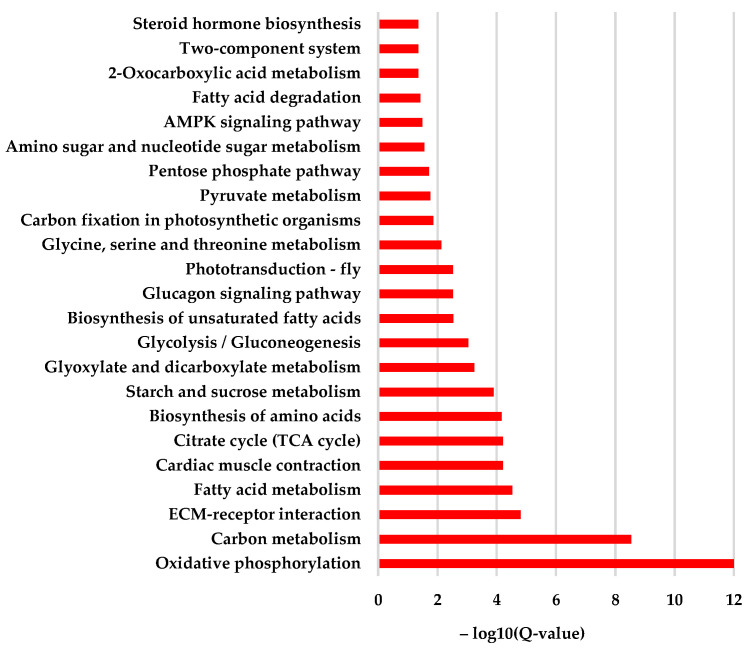
KEGG enrichment of the significantly different expressed genes in pupal bees caused by cystatin. The result showed that the cystatin injection mainly interfered with the metabolism of carbon, fatty acids, amino acids, starch, sucrose, and energy production-related processes such as oxidative phosphorylation, citrate cycle (TCA cycle), etc. In addition, the AMPK signal pathway, ECM–receptor interaction, and steroid hormone biosynthesis were also enriched.

**Figure 8 animals-13-03660-f008:**
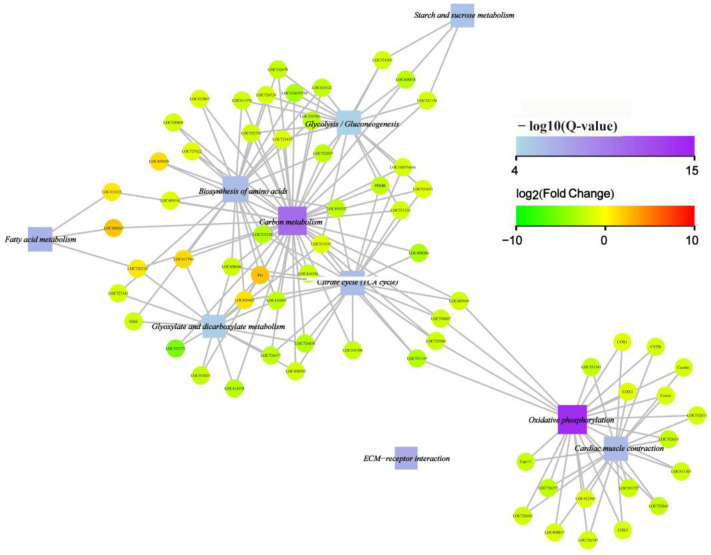
KEGG-enriched functions and gene interaction network. The circle nodes mean the specific gene; its color reflects the expression of the gene with green meaning downregulated and red upregulated. The square nodes mean the enriched functions. The relevancy between the gene and the enriched functions is line-connected. The size of the square nodes is positively related to the number of genes connected. The color of the square nodes is related to the significance of enrichment (−log10 of Q-value): the higher the enrichment, the deeper the color (the top 10 enriched functions and involved genes were selected for interaction network construction).

**Figure 9 animals-13-03660-f009:**
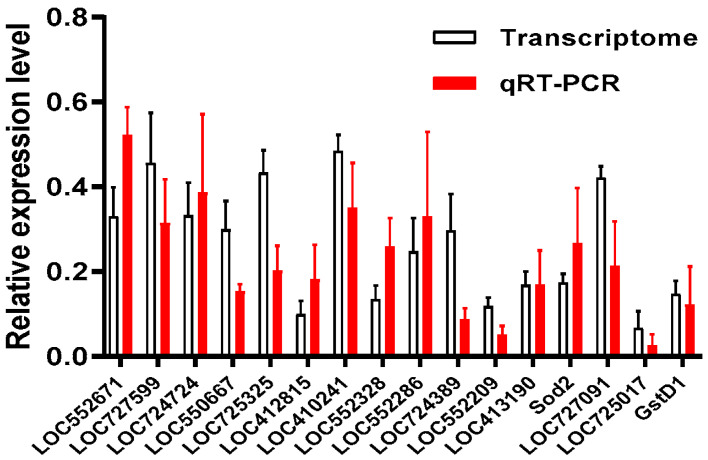
qRT-PCR validation of the partial differentially expressed unigenes (DEGs) in honeybee larvae. qRT-PCR: relative expression of selected genes determined by qRT-PCR (red column). Transcriptome: relative expression of selected genes determined by transcriptome analysis (white column). The expression of genes selected from KEGG enrichment in the larvae of the Cys200 group was downregulated when compared with expression in the larvae of the PBS group via the qRT–PCR method, which was identical to their corresponding expression levels shown in RNA-seq analysis (independent *t*-test, *p* > 0.05).

## Data Availability

The data presented in this study are available on request from the corresponding author.

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
