# Peer review of "Salivary Cystatin-L2-like of Varroa destructor Causes Lower Metabolism Activity and Abnormal Development in Apis mellifera Pupae"

_animals, 2023, doi:10.3390/ani13233660_

Round 1

Reviewer 1 Report

Comments and Suggestions for Authors

Zhou and co-authors purified a salivary protein and injected the protein into the pupae to analyze the mortality, body weight, and gene expression. This is a well-designed experiment. I have only a few minor concerns:

(1) line 19: there is no evidence to support the mite saving the bee's nutrients for itself. This is pure speculation.

(2) line 122: are the 30 mites pooled for qPCR?

(3) How many pupae were included in each group?

(4) In Figure 1, please explain how does df=4 was calculated. In my opinion, df=4 has no statistical power.

(5) In Figure 4, One-way ANOVA is not a correct method to test on a box plot. Additionally, have the multiple comparisons been adjusted?

(6) Figure 6 needs to be improved. I can barely read the text.

Author Response

Comments and Suggestions for Authors

Reviewer 1

Zhou and co-authors purified a salivary protein and injected the protein into the pupae to analyze the mortality, body weight, and gene expression. This is a well-designed experiment. I have only a few minor concerns:

  • line 19: there is no evidence to support the mite saving the bee's nutrients for itself. This is pure speculation.

Response: the sentence in line 19 is “The downregulation could save nutrient and energy for V. destructor to maximize its reproduction potential, implying that Varroa destructor could manipulate the metabolism of host bees through the injected salivary secretion”. We got the result that the cystatin down regulated the metabolism of carbohydrates, fatty acids, amino acids and ATP production of white eyed pupae bees in the study. I agree with the comment that there is no direct and strong evidence to support the conclusion that the mite saving the bee's nutrients for itself. Instead of directly drawing the conclusion that the mite saving the bee's nutrients for itself, we here used the words “could” and “implying” to weaken the strength of the expression. Thereby, we think the expression here is acceptable in general. Thanks!

(2) line 122: are the 30 mites pooled for qPCR?

Response: Yes, we pooled the 30 mites from one colony to be as one biological repeat in the study, and total 90 mites from three colonies formed three biological repeats, which were used for following RNA extraction and qPCR.

(3) How many pupae were included in each group?

Response: Total 96 pupae were included in each group. And we already indicate the information in the sentence “A total of 96 pupae on four 24-well plates for each group were prepared” in part of “2.5 Cystatin Injection to Honeybee Pupae”. Thanks!

(4) In Figure 1, please explain how does df=4 was calculated. In my opinion, df=4 has no statistical power.

Response: Figure 1 is the result of cystatin gene expression in mites in reproductive phase and dispersal phase. As state in above point (2), three biological repeats samples each pooled with 30 mites in reproductive phase and dispersal phase respectively were used in the study. Independent t-test were used to analyze the difference between the samples from the mites in reproductive phase and the mites in dispersal phase. In this case, the value of the df is equal to the “Number of samples from the reproductive phase minus 1” add with “the Number of samples from the dispersal phase minus 1”. That is the way of df=4 calculated, which is also the data produced by the software of GraphPad when we analyzed the data. Thanks!

(5) In Figure 4, One-way ANOVA is not a correct method to test on a box plot. Additionally, have the multiple comparisons been adjusted?

Response: Figure 4 showed the effect of different dosage of cystatin on the weight of newly emerged bees. Data of pupa bees from five groups of CK, PBS, Cys50, Cys100, and Cys200 were used in this part. Here only one factor of “dosage of cystatin” was involved, Group CK and PBS are as the control. Thereby, One-way ANOVA (Single factor analysis of variance) is suitable for analyzing the difference among the five groups.

In addition, we performed the Tukey's post hoc multiple comparisons test after the One-way ANOVA analysis. The detail result of the multiple comparisons refer the following table, from which the different letters above each bar in Figure 4 are come from based on the data of the Adjusted P Value. And we add the “Tukey's post hoc multiple comparisons test” into the part of 2.10 Statistical Analysis, Thanks!

Tukey's multiple comparisons test

Mean Diff.

95.00% CI of diff.

Below threshold?

Summary

Adjusted P Value

CK vs. PBS

-0.03558

-4.668 to 4.597

No

ns

>0.9999

A-B

CK vs. Cys50

14.35

8.020 to 20.68

Yes

****

<0.0001

A-C

CK vs. Cys100

19.63

9.501 to 29.77

Yes

****

<0.0001

A-D

CK vs. Cys200

27.49

16.13 to 38.85

Yes

****

<0.0001

A-E

PBS vs. Cys50

14.38

8.001 to 20.77

Yes

****

<0.0001

B-C

PBS vs. Cys100

19.67

9.503 to 29.84

Yes

****

<0.0001

B-D

PBS vs. Cys200

27.53

16.14 to 38.92

Yes

****

<0.0001

B-E

Cys50 vs. Cys100

5.286

-5.758 to 16.33

No

ns

0.6804

C-D

Cys50 vs. Cys200

13.14

0.9641 to 25.32

Yes

*

0.0273

C-E

Cys100 vs. Cys200

7.857

-6.667 to 22.38

No

ns

0.5705

D-E

(6) Figure 6 needs to be improved. I can barely read the text.

Response: Because of limited space, we could not display the Figure 6 clearly. We have already supplied the original Figure file with enough resolution so that a better Figure 6 could replace the present one. Thanks!

Reviewer 2 Report

Comments and Suggestions for Authors

The paper titled 'Salivary Cystatin-L2-like of Varroa destructor Causes Reduced Metabolic Activity and Abnormal Development in Apis mellifera Pupae' offers valuable insights into the impact of the cystatin protein found in the salivary secretions of Varroa destructor on the physiology and survival of honeybees. The experimental design is sound, and the presentation of results is clear. However, there are a few minor issues that warrant the authors' attention.

The research involves injecting cystatin into honeybee pupae, but the rationale for selecting the specific doses is not well-explained. It would be beneficial if the paper provided insight into whether these doses reflect realistic amounts of the protein that mites typically inject. Overall, the paper requires minor corrections for enhanced clarity.

Comments on the Quality of English Language

Overall, the English writing is clear; however, minor corrections are needed to enhance the understanding of certain sentences

Author Response

Reviewer 2

The paper titled 'Salivary Cystatin-L2-like of Varroa destructor Causes Reduced Metabolic Activity and Abnormal Development in Apis mellifera Pupae' offers valuable insights into the impact of the cystatin protein found in the salivary secretions of Varroa destructor on the physiology and survival of honeybees. The experimental design is sound, and the presentation of results is clear. However, there are a few minor issues that warrant the authors' attention.

The research involves injecting cystatin into honeybee pupae, but the rationale for selecting the specific doses is not well-explained. It would be beneficial if the paper provided insight into whether these doses reflect realistic amounts of the protein that mites typically inject. Overall, the paper requires minor corrections for enhanced clarity.

Response: Thanks for the positive comments on our manuscript. The suggestion on the revision of some sentences and expression in the multiple places of the manuscript is really helpful for improving the quality. And we accordingly revise them one by one, the specific modification content refer the revised manuscript, Thanks!

As to the rationale for selecting the specific doses of cystatin in the study, we did not well explain it in manuscript. This is good suggestion. Doses selecting mainly refer a previous report in which the salivary secretion of Varroa destructor was collected, and injected into the honeybee for determining the secretome and toxicity to honeybee, they select a dosage of 0.2μl of Varroa mite saliva with concentration of 2.16 μg/μl (which corresponding to the dosage of 432 ng saliva injected), and the injected worker L5 larvae of A. mellifera could developed into adults with deformed wings, which is similar to the symptom of two mites infection in the lab. (Zhang, Y.; Han, R. Insight Into the Salivary Secretome of Varroa destructor and Salivary Toxicity to Apis cerana. Journal of economic entomology 2019, 112, 505-514, doi:10.1093/jee/toy224). Thereby, we start 400 ng doses in our pre-test, but found that all the pupae bees died quickly, which impede to obtain the detail effect of cystatin on honeybee. We then make a series two times of dilution, and found the present doses of cystatin caused the different level of lethal rate, and were thereby selected in the study.

In according with the suggestion, we add this sentence “We tested injection of 400 ng of cystatin to pupae bees referring the previous report injecting Varroa mite saliva to both A. mellifera and A. cerana [38]. But all the test pupae bee died quickly. Then, three round of two times dilution of cystatin were tested and showed the different level of lethal rate. Therefore……” to the manuscript for explaining the dose rationale. Thanks!

As to the second suggestion to discuss whether these doses reflect realistic amounts of the protein that mites typically inject. This is really a valuable but challenging suggestion. We say it is valuable because such a discussion could compare the doses with the realistic amounts mites injected that could deepen our understanding to the result. And the same time, we say the suggestion is challenging because nobody know for sure how much the amounts of mites saliva was injected into the host bees. Firstly, the salivary secretion along with the feeding activity last for the whole pupae development stage. It was not injected all at once. Secondly, the feeding and salivary secretion activity to the host bees was not just performed by the mother mite, her offspring might in different developmental stage also feed the same pupae bee even through the same hole on the cuticle made by the mother mite. Thirdly, the mite salivary secretion injected into host bees could go back to the mite through their feeding activity, which make the determination of mite saliva amount complicated. It may be just because of this, we found no study involved in determining the realistic amount of mite saliva typically injected so far.

However, the above data that honeybees injected 432 ng of mite saliva has similar symptom to two mite’s infection in the lab still give us a good reference. We compare our dose with this data in discussion, and try to establish a connection of our dose with the realistic mite infection. The following sentence “Moreover, previous report showed that injection of 432 ng of Varroa mite saliva caused the emerged bees symptom of deformed wings, which was equal to the symptom caused by two Varroa mites in lab [38]. While in this study, injection of the lowest dosage of 50 ng cystatin could cause over 50% of pupae bees died, exhibiting much stronger effects than the Varroa mite itself or the Varroa mite saliva as whole on honeybee” was thus added to the first paragraph in discussion. Thanks!